# Molecularly Targeted Therapy towards Genetic Alterations in Advanced Bladder Cancer

**DOI:** 10.3390/cancers14071795

**Published:** 2022-04-01

**Authors:** Jonathan Thomas, Guru Sonpavde

**Affiliations:** 1Lank Center for Genitourinary Oncology, Dana-Farber Cancer Institute, Boston, MA 02215, USA; jdthomas@bidmc.harvard.edu; 2Division of Medical Oncology, Department of Medicine, Beth Israel Deaconess Medical Center, Boston, MA 02215, USA

**Keywords:** muscle-invasive bladder cancer, gene fusions, targeted therapy, fibroblast growth factor receptor, erdafitinib, ErbBreceptor, PI3K, Akt, mTOR, MAPK, chromatin remodeling, cell cycle regulation, DNA damage repair

## Abstract

**Simple Summary:**

In recent years, there have been several advances in the care of advanced bladder cancer, highlighted by the addition of immune checkpoint inhibitors to the care of advanced disease. Despite these advances, there is a need for further improvement; the morbidity and mortality associated with advanced bladder cancer remain high. With the recent incorporation of advanced molecular techniques, there is more clarity regarding key genetic alterations of the disease. Therapies directed at specific genetic aberrations in bladder cancer provide both proven and potential paths forward. This review discusses the key targetable genetic aberrations and summarizes the current status of targeted therapies in muscle-invasive bladder cancer.

**Abstract:**

Despite the introduction of immune checkpoint inhibitors and antibody–drug conjugates to the management of advanced urothelial carcinoma, the disease is generally incurable. The increasing incorporation of next-generation sequencing of tumor tissue into the characterization of bladder cancer has led to a better understanding of the somatic genetic aberrations potentially involved in its pathogenesis. Genetic alterations have been observed in kinases, such as FGFRs, ErbBs, PI3K/Akt/mTOR, and Ras-MAPK, and genetic alterations in critical cellular processes, such as chromatin remodeling, cell cycle regulation, and DNA damage repair. However, activating mutations or fusions of *FGFR2* and *FGFR3* remains the only validated therapeutically actionable alteration, with erdafitinib as the only targeted agent currently approved for this group. Bladder cancer is characterized by genomic heterogeneity and a high tumor mutation burden. This review highlights the potential relevance of aberrations and discusses the current status of targeted therapies directed at them.

## 1. Introduction

Urothelial carcinoma includes tumors of the bladder as well as the renal pelvis and ureters. Bladder cancer is the location of around 90% of urothelial carcinoma (UC) [1]. It is the tenth most common cancer worldwide [2]. There are approximately 570,000 new cases of bladder cancer diagnosed annually, with a male-to-female ratio of 4:1 [2]. In the United States, there are 81,000 new cases and 17,000 deaths from bladder cancer projected for 2022 [3].

Advanced or metastatic bladder cancer represents 4% of newly diagnosed bladder cancer. The vast majority of new cases (~75%) have non-muscle-invasive disease. However, among those who present with initially muscle-invasive bladder-confined disease, ~50% progress to metastatic disease [1]. In advanced or metastatic disease, first-line treatment with cisplatin-based chemotherapy has a median overall survival of ~15 months, while those who are cisplatin-ineligible have suboptimal outcomes (~9 months) [4,5]. Recent advances have seen the addition of programmed cell death protein 1/programmed cell death ligand 1 (PD1/L1) inhibitors in the management of metastatic disease in patients who are post-platinum or cisplatin/platinum-ineligible [6,7,8,9]. Despite these advancements, the objective response rate with PD1/L1 inhibitors is 21–27%, and the median survival is 8–11 months [6,9]. PD1/L1 inhibitors are also being used as maintenance therapy following platinum-based therapy in patients with responding or stable disease [10]. In addition, antibody–drug conjugates (ADCs), enfortumab vedotin (EV), and sacituzumab govitecan (SG) have emerged as valuable additions to the therapeutic armamantarium [11,12,13]. Despite these advances, metastatic disease is generally incurable.

As next-generation genomics technologies are being employed, additional potentially targetable pathways are being revealed [14,15,16]. Erdafitinib, which targets the family of fibroblast growth factor (FGFR) tyrosine kinase receptors, is the first targeted therapy to be FDA-approved for the treatment of advanced urothelial carcinoma following platinum-based chemotherapy and harboring activating mutations or fusions in *FGFR2* or *FGFR3* [17]. In addition to the FGFRs, several other potentially targetable genetic alterations involving a variety of cellular functions have been implicated in bladder cancer, including the ErbB receptors, the PI3K/Akt/mTOR pathway, the RAS-MAPK signaling pathway, chromatin remodeling, cell cycle regulation, and DNA damage repair (Figure 1). In this review, we summarize the potentially actionable somatic genomic alterations in muscle-invasive and advanced urothelial carcinoma and highlight ongoing efforts to develop therapeutics directed at these alterations.

## 2. FGFRs

There are four fibroblast growth factor receptors (FGFR1–4) [18]. FGFRs are tyrosine kinase receptors that bind at least 18 fibroblast growth factors (FGFs). Once binding fibroblast growth factors, the FGFRs dimerize and become activated via phosphorylation of their cytoplasmic domains [19,20,21]. Activation leads to signaling via a variety of pathways including PLCγ1, RAS-MAPK, PI3K, and STATs [22]. These pathways regulate many functions including cellular migration, proliferation, and differentiation [22].

FGFR3 dysregulation via mutation, overexpression, or both have been noted in 54% of invasive urothelial carcinomas (UCs) [23]. Interestingly, mutations in FGFR3 are more frequent (~80%) in non-invasive UCs and in upper tract UC (~1/3), which are enriched for luminal papillary gene expression subtype tumors with poor immune infiltration [14,15,16]. Meanwhile, FGFR3 mutations have been implicated in 5–20% of muscle-invasive bladder cancer (MIBC) [14,23,24]. These aberrations are commonly point mutations in the extracellular region leading to ligand-independent dimerization, activation, and signaling [15,25,26]. Invasive tumors are more likely to have upregulation of wild-type FGFR3 [15]. Overexpression of the wild-type FGFR3 may lead to ligand-independent dimerization and activation [15]. Ligand-dependent mechanisms via increased levels of fibroblast growth factors may also contribute to tumor development, although the therapeutic actionability is unclear [27].

FGFR3-TACC gene fusions are also relevant to MIBC and were identified more frequently in younger patients, never smokers and those with Asian ethnicity [28]. TACC3 is involved in the stability and organization of the mitotic spindle [29]. FGFR3-TACC3 fusions are formed via tandem duplications on chromosome 4p16 [30]. FGFR3-TACC3 fusions are relatively rare, with a frequency of 2–3% in MIBC [31,32,33]. Regarding pathogenesis of the fusion, in glioblastoma, the FGFR3-TACC3 fusion has been shown to lose a regulatory site for miR-99a, an inhibitor of FGFR3, leading to enhanced expression of the gene fusion [30]. Furthermore, a coiled-coil domain on TACC3 increases the activity of FGFR3 via phosphorylation of its tyrosine kinase domains [34]. The FGFR3-TACC3 fusion also results in alterations of TACC3 function, which can result in mitotic defects and aneuploidy [35,36,37].

FGFR1 aberrations have been less frequently studied than FGFR3. The prevalence of FGFR1 genomic alterations was noted to be 7–14% [32,38]. FGFR1 and subsequent MAPK activation have been noted to promote proliferation and survival as well as induce epithelial-to-mesenchymal transition (EMT) [39,40].

Resistance to FGFR inhibition can develop due to upregulation of bypass pathways, such as PI3K-AKT, RAS-MAPK, and STAT [41]. Gatekeeper mutations of the FGFR binding domain have also been associated with resistance, e.g., FGFR1 V561M, FGFR2 V564F/I, FGFR3 V555M, and FGFR4 V550E/L [41,42,43,44]. Specific FGFR inhibitors may target these gatekeeper mutations; e.g., Debio 1347 was effective against FGFR2 V564I, and futibatinib was active in FGFR inhibitor refractory patients [45,46].

### 2.1. First-Generation FGFR Inhibitors

While the individual mechanisms of actions are nuanced beyond the scope of this article, the majority of FGFR inhibitors listed function by binding near the adenine-binding site of the receptor tyrosine kinase [47]. First-generation and promiscuous FGFR tyrosine kinase inhibitors have been evaluated as monotherapy or in combination with cisplatin-based chemotherapy or PD1 inhibition in unselected patients, e.g., dovitinib, nintedanib, and lenvatinib [48,49,50]. Nintedanib did not improve pathologic complete response in combination with neoadjuvant cisplatin-based combination; however, there was an improvement in PFS and OS [49]. Lenvatinib inhibits VEGFRs in addition to FGFRs and has demonstrated promising activity in combination with pembrolizumab as first-line therapy in unselected patients, which led to an ongoing phase III trial comparing pembrolizumab vs. lenvatinib plus pembrolizumab in platinum-ineligible or cisplatin-ineligible patients with PD-L1 high-expressing tumors (NCT03898180) [50].

### 2.2. Erdafitinib

Several potent and FGFR-specific second-generation therapeutics have been developed. Erdafitinib is an orally bioavailable selective and potent inhibitor of FGFR1–4 [17]. Erdafitinib was evaluated in an open-label, single-arm, phase II trial of 99 patients with locally advanced and unresectable or metastatic urothelial carcinoma with FGFR2 or 3 mutations or fusions who had progressed on prior platinum-based chemotherapy [51,52]. Notably, the dose of erdafitinib was guided by off-cancer on-target induction of hyperphosphatemia by inhibition of FGFR1. Those with an absence of hyperphosphatemia ≥5.5 nmol/L at ~2 weeks were dose-escalated to 9 mg once daily in the absence of other prohibitive toxicities. The primary endpoint of objective response rate (ORR) was achieved in 40% of patients (3% complete, 37% partial). The median progression-free survival (PFS) was 5.5 months, and the median overall survival (OS) was 11.3 months after a median follow-up of 24 months [52]. The response rate was not affected by visceral metastasis. The ORR appeared higher in those with FGFR mutations vs. fusions (49% and 16%, respectively). EGFR, CCND1, and BRAF alterations in baseline ctDNA were associated with poor outcomes. Common grade 3 or 4 toxicities included hyponatremia (11%), stomatitis (10%), and asthenia (7%). Notable FGFR-inhibitor class-specific toxicities included hyperphosphatemia, which was present in 77% of patients, and ocular toxicities (10% with grade 3 or higher), such as retinal pigment detachment or central serous retinopathy. Based on this trial, erdafitinib was granted accelerated FDA approval in April 2019 for the treatment of locally advanced or metastatic urothelial carcinoma with FGFR2 or FGFR3 genomic alterations that have progressed on platinum-based chemotherapy.

Since erdafitinib’s FDA approval, it remains unclear what the appropriate sequence of second-line therapies is between immune checkpoint inhibitors or erdafitinib in those with FGFR-activating alterations [53]. FGFR-altered UC has been suggested to be less likely to respond to immunotherapy, due to reduced T-cell infiltration in FGFR-altered UC [54]. Indeed, in the above phase II trial that evaluated erdafitinib, of 19 patients who received a PD1/L1 inhibitor previously, only one responded. However, another retrospective study assessing response rates in relation to FGFR3 mutation status, including two key immunotherapy trials (IMVigor 210 and CheckMate 275) that administered post-platinum atezolizumab or nivolumab, noted similar response rates in patients with and without FGFR3 mutations [16,53,55]. The association of FGFR3-mutant UC with lower T-cell infiltration may be counteracted by lower stromal transforming growth factor (TGF)-β, which is a driver of epithelial–mesenchymal transition and therefore a source of resistance against the activity of PD1/L1 inhibition. Toxicity profiles also differ between immunotherapy and FGFR inhibition. There is a current randomized phase III trial comparing the benefit of erdafitinib versus PD1/L1 inhibitor immunotherapy or chemotherapy in metastatic or surgically unresectable UC with progression following prior therapy (NCT03390504) (see Table 1). In addition, there is an ongoing randomized phase II trial comparing erdafitinib alone vs. erdafitinib with anti-PD-L1 therapy (cetrelimab) as first-line therapy in patients with metastatic UC who are cisplatin-ineligible (NCT03473743). Preliminary results on interim analysis showed a promising increase in ORR with combination therapy, and accrual continues [56].

### 2.3. Investigational Specific FGFR Inhibitors

In addition to erdafitinib, several other inhibitors of FGFRs are being evaluated. Rogaratinib, an FGFR1–4 inhibitor, was evaluated in an open-label, randomized, phase II/III study comparing rogaratinib with chemotherapy (docetaxel, paclitaxel, or vinflunine) in locally advanced or metastatic UC patients who had received platinum-based chemotherapy and had tumors overexpressing FGFR1 or 3 mRNA (NCT03410693) [57]. Unfortunately, the trial was stopped early due to futility for improved outcomes with rogaratinib. Similar overall response rates were noted with rogaratinib versus standard chemotherapy (19.5%/19.3%, respectively). Grade 3 or 4 toxicities were present in 47% of patients receiving rogaratinib and 56% of patients receiving chemotherapy. Of note, on subsequent analysis, the overall response rate was higher in patients with FGFR3 DNA alterations (52.4% with rogaratinib and 26.7% with chemotherapy), suggesting that selecting patients based on gene expression may not optimally enrich for tumors driven by FGFR. There is an ongoing phase Ib/II study of the combination of rogaratinib with atezolizumab in patients with advanced/metastatic UC who are first-line cisplatin-ineligible and exhibit FGFR gene overexpression (NCT03473756) [58]. Preliminary results in 26 patients reported a disease control rate of 83%.

Infigratinib (BGJ398), an orally bioavailable, selective FGFR1–3 inhibitor, was evaluated in a single-arm, phase II trial of 67 patients with metastatic UC who had either progressed on or were intolerant of platinum-based chemotherapy and harbored tumors FGFR3 genetic alterations [59]. The study noted 64.2% of patients with disease control, ORR of 25.4%, and a median PFS of 3.75 months. There is an ongoing phase III, double-blind, randomized, placebo-controlled trial assessing infigratinib in the adjuvant setting in patients with high-risk muscle-invasive UC at risk for recurrence (NCT04197986). A phase I trial is evaluating infigratinib in the neoadjuvant setting for cisplatin-ineligible patients with FGFR3-activating genomic alterations who are candidates for radical cystectomy (NCT04972253).

Pemigatinib (INCB054828) is a selective inhibitor of FGFR1–3 [60]. Preliminary results in the metastatic setting demonstrated efficacy with an overall response rate of 25% in a cohort of 64 patients [61]. There are ongoing phase II trials in two settings in UC: adjuvant (NCT04294277) and metastatic or surgically unresectable (NCT02872714).

Other investigational small-molecule oral FGFR inhibitors undergoing evaluation include derazantinib, futibatinib, and AZD4547 (Table 1) [62,63]. In contrast to the combination of erdafitinib and cetrelimab, the combination of durvalumab and AZD4547 did not appear to improve efficacy following platinum-based chemotherapy in the biomarker-guided BISCAY trial [64]. Vofatamab (B-701), a monoclonal antibody that prevents activation of both wild-type and mutated FGFR3, was evaluated as a single agent or in combination with docetaxel in a phase Ib/II trial in patients with relapsed or refractory metastatic UC with *FGFR* mutations or fusions (NCT02401542) [65]. The ORRs of vofatamab and the combination were 4.8% and 19.0%, respectively. Vofatamab was also being evaluated in another phase Ib/II trial in combination with pembrolizumab in metastatic UC in patients who had progressed on chemotherapy (NCT03123055) [66]. Interestingly, in preliminary results, the response to vofatamab did not correlate with FGFR3 mutation or fusion (42.9% with mutation/fusion and 33.3% in FGFR-wildtype patients) [66].

## 3. ErbB Receptors

The ErbB group of receptor tyrosine kinases (RTKs) is composed of four receptors: EGFR (ErbB-1/HER1), ErbB-2 (neu, HER2), ErbB-3 (HER3), and ErbB-4 (HER4) [67]. ErbB RTKs are activated by binding EGF-family growth factors via paracrine or autocrine secretion [68]. Ligand binding induces receptor dimerization followed by activation of the intracellular tyrosine kinase domain. Numerous signal transduction pathways can then be initiated, including activation of the Ras-MAPK [69,70] or the PIK3CA pathway [71].

Within the ErbB family, mutations or amplifications in EGFR, ErbB-2, and ErbB-3 have been associated with MIBC [31]. EGFR aberrations are present in 6–14% [31,38,72] of MIBC, while ErbB-2 mutations are present in 6–23%, and ErbB-3 mutation are present in 6%. Overactivation of ErbB receptors increases activation of signaling pathways, promoting cell proliferation and survival [73]. In addition, ErbB receptor aberrations have been associated with increased chromosome instability [74].

### 3.1. ErbB1 Receptor Inhibitors

There have been multiple clinical trials of ErbB receptor inhibitors in patients with MIBC. Gefitinib, an EGFR-TKI inhibitor, was evaluated in combination with first-line chemotherapy in patients with advanced or metastatic UC in a phase II trial [75]. There was no significant difference in time to progression in patients randomized to receive gefitinib when compared to chemotherapy alone. Cetuximab, an EGFR-TKI inhibitor, was also evaluated in a phase II trial in patients with advanced UC. There were no improvements in outcomes in patients treated with cetuximab; however, there was an increase in adverse events [76]. Of note, EGFR mutational testing was not performed in patients in either of the trials mentioned above.

### 3.2. ErbB2 (HER2) Inhibitors

Trastuzumab, a monoclonal antibody targeting ErbB-2, has an established role in gastric and breast cancer. It was evaluated in locally advanced or metastatic UC in a multicenter randomized phase II first-line study [77]. Her2 protein overexpression by immunohistochemistry was a part of the eligibility criteria. Patients were randomized to chemotherapy (gemcitabine with cisplatin or carboplatin) alone or chemotherapy plus trastuzumab. Unfortunately, there was no significant difference in progression-free survival or overall survival. Notably, 563 patients were screened to randomize 61 patients, highlighting the challenges of conducting trials in biomarker-selected patients. Additional trials assessing the effectiveness of trastuzumab are ongoing (Table 2). For example, the “MyPathway” study (NCT02091141) is a multi-basket study, of which one study arm is assessing trastuzumab with pembrolizumab in ErbB-2 positive metastatic urothelial carcinoma [78].

Lapatinib, a tyrosine kinase inhibitor, which inhibits both EGFR and ErbB-2, was evaluated in a phase III trial of patients with metastatic UC and EGFR or Her2 protein expression who had received chemotherapy and did not have progressive disease [79]. Patients were randomly assigned to lapatinib or placebo. The results were disappointing, and there were no improvements in outcomes in patients treated with lapatinib.

While inhibition of Her2 is an established treatment in breast and gastric cancer, inhibition of ErbB receptors has not demonstrated similar effectiveness in MIBC and advanced UC to this date. The trials mentioned above have either not screened for Her2 status or have used FISH and IHC to identify patients overexpressing ErbB2. These studies may not have adequately selected patients who may benefit from ErbB inhibition. Kiss et al. performed a comprehensive analysis of Her2 alterations at the DNA, RNA, and protein level [80]. They reported a significant proportion of ErbB2 amplifications without overexpression of ErbB2. They identified instances of ErbB2 overexpression without amplification of ErbB2. They suggest that gene amplification is not the only factor increasing high ErbB2 expression in MIBC. The presence of single-nucleotide variations (SNVs) in the extracellular domain of the ErbB2 receptor resulted in a lower affinity of antibody binding, which may lead to falsely low levels of expression reported via IHC and may also affect the binding of ErbB2 inhibitors. They also noted that when compared to gastric and breast cancer, bladder cancer has a relatively high rate of genomic alterations including mutations and amplification [31]. The high rate of genomic alterations potentially reduces the likelihood that an individual alteration such as ErbB2 is a significant oncogenic driver even when it is overexpressed. Notably, there were cases in which ErbB2 was overexpressed in the absence of other known oncogenic alterations. In these cases, it is more likely that ErbB2 has oncogenic relevance. All in all, Kiss et al. suggest an algorithm incorporating the presence of IHC for ErbB2 as well as detection of gene amplification for ErB2, somatic mutations in ErbB2, and the presence of other oncogenic gene amplifications as a more targeted means of assessing patients who may benefit from ErbB2 inhibition. Their algorithm requires clinical validation. A more refined targeted approach may improve the outcomes of ErbB inhibitors in MIBC.

Another potential therapeutic strategy is to target ErbB2 with antibody–drug conjugates (ADCs). ADCs combine a selective monoclonal antibody with a potent cytotoxic agent. The ADC, traztuzumab deruxtecan, is being combined with nivolumab in a phase Ib, two-part open-label study in patients with advanced/metastatic urothelial carcinoma who have progressed on prior platinum chemotherapy and have tumor HER2 protein expression based on immunohistochemistry [81]. Of the 30 patients with high expression 2+ or 3+ of Her2, the overall response rate was 36.7%. Activity appeared higher in those with 3+ expression but was also observed in a small group of patients with 1+ expression. Toxicities were consistent with prior reports, and interstitial lung disease/pneumonitis occurred in 23.5% of patients, with one event leading to death. RC48-ADC is another ADC targeting ErbB2, which is being evaluated in combination with toripalimab, an anti-PD-1 antibody, in a phase Ib/II study of patients with advanced/metastatic UC who are cisplatin-ineligible or have progressed on one line of standard chemotherapy [82]. While ErbB2 status was assessed with immunohistochemistry, expression was not required for enrollment. Preliminary results of 32 patients reported an overall response rate of 75%. Of note, 56% of patients enrolled had upper tract UC. Further evaluation of RC48-ADC appears warranted. All in all, early phase clinical trials of ADCs directed at ErbB2 have had promising results but will require validation. Additional trials of RC48-ADC, trastuzumab deruxtecan, and ado-trastuzumab are ongoing (Table 2).

## 4. PI3K/Akt/mTOR Pathway

The PI3K/Akt/mTOR pathway is an intracellular pathway involved in the regulation of cell cycle progression, cellular growth, angiogenesis, and apoptosis [83,84]. Phosphoinositide 3-kinase (PI3K) is activated by means of a variety of growth factors binding their receptors, including the FGFR and ErbB families of receptors [85,86]. Activated PI3K promotes activation of Akt1 via PIP3 [87]. Activated Akt1 inhibits the tuberous sclerosis complex (TSC) of TSC1 and TSC2. When activated, TSC inhibits Rheb. When Rheb is no longer inhibited by TSC, Rheb is then able to activate mTOR. Activated mTOR proceeds to promote cell cycle progression and cellular growth through interaction with multiple effectors [84]. Another relevant regulator of the PI3k/Akt/mTOR pathway is PTEN. PTEN prevents activation of Akt1 by dephosphorylation of PIP3 [88]. In addition, PTEN regulates cell motility and chemotaxis [89]. There are several other regulators of this key pathway that are beyond the scope of this review.

Genetic alterations in the PI3K/Akt/mTOR pathway are common in urothelial carcinoma. PI3KCA, the catalytic subunit of PI3K, has been noted to be altered in 20–26% of advanced urothelial carcinomas [31,38]. Different activating mutations of PIK3CA vary in their degree of signaling activation and ligand independence [90]. Certain hotspot point mutations in UC, including E545G, are noted to have the highest effects. Akt genetic alterations have been noted in 6% of advanced urothelial carcinomas [38]. The Cancer Genome Atlas Network (TCGA) noted AKT mRNA overexpression in 10% of cases [31]. In addition to its role in the PI3K/Akt/mTOR pathway, increased Akt activity has been shown to reduce apoptosis via resistance to pro-apoptotic ligands [91]. Inactivating mutations or deletions of TSC1 or TSC2 were noted in 6–11% of cases [31,38,92]. The tumor suppressor PTEN was inactivated or deleted in 3–13% of cases [31,38]. Loss of PTEN is also associated with inactivation or deletion of p53 and has been associated with worse patient outcomes and more aggressive tumor features [93,94,95]. Taken together, genetic alterations within the PI3K/Akt/mTOR pathway have been noted in 42% of cases [31]. PI3K inhibition has been shown to have immunomodulatory effects on immune cells including CD8 and dendritic cells [96]. This provided the rationale for the combination of pan-PI3K inhibition and immune checkpoint inhibition, which exhibited significant antitumor effects in advanced UC with or without activated PI3K pathway by creating an immunostimulatory tumor milieu. Considering the frequency of alterations to the pathway in MIBC, various components of the pathway have understandably been the focus of several targeted therapies.

Despite their biological rationale, inhibitors of the PI3K/Akt/mTOR pathway have not had promising results in clinical trials to date. The PI3K/Akt/mTOR pathway has significant cross-talk with the Ras-ERK pathway [97,98]. There is also some redundancy in the activation of downstream effectors between the two pathways [98,99,100]. Due to the inter-pathway regulation, inhibition of the PI3k/Akt/mTOR can result in increased activity of the MAPK pathway and thus serve as a cellular escape mechanism [101]. In addition, inhibition of the PI3K/Akt/mTOR pathway has been shown to lead to increased expression and phosphorylation of multiple receptor tyrosine kinases [102]. This increased expression may further reduce the effectiveness of inhibition of the PI3k/Akt/mTOR pathway.

### 4.1. PI3K Inhibition

Buparlisib (BKM120) is an orally bioavailable class I PI3K inhibitor which inhibits both wild-type and mutated PI3K [103]. Buparlisib was evaluated in a single-arm, open-label phase II study in patients with locally advanced or metastatic, platinum-refractory UC [104]. The primary endpoint was two-month progression-free survival (PFS). The initial cohort of 16 patients did not select for genetic alterations in the PI3K/Akt/mTOR pathway. The two-month PFS was 54%. In a subsequent genetically selected cohort, patients were selected for PIK3CA, Akt1, or TSC1 mutations. There were only four patients in the genetically selected cohort before study termination by the sponsor. None of the four patients were progression-free at 8 weeks. There were significant adverse events requiring dose reduction in 38% of the cases and causing two patients to withdraw early.

### 4.2. AKT

MK-2206, an AKT inhibitor, has been primarily studied in breast cancer in phase I and II trials without promising results [105]. Pre-clinical studies have shown MK-2206 potentiates the activity of cisplatin in urothelial carcinoma; however, this has not been further evaluated in clinical trials [106].

### 4.3. mTOR

Rapamycin, also called sirolimus, was the first mTOR inhibitor discovered [107]. It has been shown to inhibit the mTORC1 complex, which is one of the downstream effector complexes of mTOR involved in the regulation of translation and cell growth [107,108]. There are now additional mTOR inhibitors including temsirolimus and everolimus [107].

mTOR inhibitors have an established role in the treatment of other cancers such as advanced kidney cancer and pancreatic neuroendocrine tumors [109,110]. Pre-clinical studies of mTOR inhibitors in urothelial cancer have demonstrated dose-dependent inhibition of proliferation in UC cell lines [111]. However, clinical trials of mTOR inhibitors in patients with UC have generally performed poorly.

Everolimus was evaluated in a single-arm, phase II trial in patients with metastatic UC who had progressed on at least one cytotoxic agent [112]. The study enrolled 45 patients and did not meet its primary endpoint. The progression-free survival at two months was 51%. However, there were two patients with partial responses. One patient had a durable complete response of over two years. Subsequent genetic analysis of that patient’s tumor genome identified a TSC1-inactivating mutation as well as a mutation in neurofibromatosis type 2 (NF2) [113].

Everolimus was evaluated in another open-label, single-arm phase II trial in patients with advanced or metastatic transitional cell carcinoma who had progressed on platinum-based chemotherapy [114]. The disease control rate at two months was 27%. This trial identified that PTEN loss was observed only in patients with progressive disease. PTEN expression was detected in all 6 patients with controlled disease and in 6 of 14 patients (43%) with noncontrolled disease. In a subsequent study analyzing archival tissue from the participants, it was found that the PTEN-deficient cells had increased Akt activation upon treatment with an mTOR inhibitor [115]. The authors suggest that the increased Akt activation upon exposure to mTOR inhibition in PTEN deficient tumors was a possible resistance mechanism to everolimus. In addition to PTEN deficiency, Akt activity may be increased in this setting because everolimus, which selectively blocks a subunit of mTORC1, may lead to increased mTORC2 activation, which is known to activate Akt [116]. Everolimus in addition to paclitaxel was trialed in a phase II trial of second-line treatment after failure of platinum-based chemotherapy in advanced UC [117]. The overall response rate was only 13%. Unfortunately, an umbrella trial evaluating everolimus and enrolling patients with advanced solid tumors that harbored TSC1/TSC2 or mTOR mutations demonstrated poor outcomes [118]. Partial responses were seen in only 2 patients of 30 patients, and there was no clear association between other genomic alterations and response. Of the two patients with response, one had upper tract urothelial carcinoma with biallelic inactivation of TSC1 and high tumor mutational burden.

Temsirolimus was evaluated in a phase II trial of patients with recurrent or metastatic UC who have received first-line chemotherapy [119]. Of the 45 patients evaluated, 48.9% had non-progression at two months. Four patients were treated for over 30 weeks. Of note, 52.8% had grade 3 or 4 toxicities, and 11 patients stopped treatment due to toxicity. For this reason, recruitment was halted in the trial. Another phase II trial of temsirolimus as second-line therapy in patients with metastatic UC was stopped early due to lack of sufficient benefit [120].

Sapanisertib, unlike the mTOR inhibitors discussed above, inhibits both mTORC1 and mTORC2 complexes [121]. By inhibiting mTORC2 as well, Akt activity is not subsequently upregulated [122]. A phase II trial of sapanisertib in locally advanced or metastatic UC with TSC1 or TSC2 mutations who have developed disease progression after platinum-based chemotherapy was terminated early for futility as well as for a high prevalence of adverse events (hyperglycemia, acute kidney injury, and elevated liver enzymes [123].

Dactolisib, (BEZ235) a PI3K and mTORC1/2 inhibitor, was evaluated in a phase II trial in UC [124]. Dactolisib demonstrated modest clinical activity, and the progression-free survival at 16 weeks was 10%. There was also considerable toxicity. The combination of durvalumab and an investigational mTORC1/2 inhibitor did not appear to improve efficacy following platinum-based chemotherapy in the BISCAY trial [64].

### 4.4. Possible Future Therapeutic Strategies Targeting the PI3K/Akt/mTOR Pathway

In light of the identified mechanisms of resistance to PI3K/Akt/mTOR inhibition, there are a few potential pathways forward. One option includes combining inhibition of the PI3K/Akt/mTOR pathway and the MAPK pathway. Inhibition of the two pathways simultaneously has been evaluated in various cancers and has been complicated by significant toxicities in most clinical trials to date [125]. However, the combination was tolerated in a more recent phase I trial with BRAF inhibition/PI3K inhibition in patients with melanoma or gastrointestinal stromal tumors [126]. There may still be a possibility of finding a tolerable combination of PI3k/Akt/mTOR inhibitor and MAPK pathway inhibitor in urothelial carcinoma.

The noted upregulation of RTKs with inhibition of PI3k/Akt/mTOR pathway raises the possibility that combining inhibition of both pathways may be more effective. Everolimus and pazopanib, a vascular endothelial growth factor (VEGF)-receptor inhibitor, was evaluated in patients with locally advanced or metastatic UC as an expansion cohort from a phase I trial of patients with advanced cancer [114]. The overall response rate was 21%. Mutations in TSC1/2 or mTOR were noted in a subset of the patients who derived a clinical benefit. Four of five patients with clinical benefit had mutations in TSC1/TSC2 or mTOR, and a fifth patient had a FGFR3-TACC3 fusion. Combined PIK3CA and FGFR inhibition was evaluated in patients with various PIK3CA-mutated solid tumors in a phase IB dose-escalation and expansion study [127]. This study noted grade 3 or 4 adverse events in 60% of patients. Interestingly, dose-limiting toxicities were only noted in 10% of cases. While the study was not powered for efficacy, the partial response rate was only 9.7%. There were no clear genetic associations with partial response after performing next-generation sequencing.

Despite the modest results of these trials, further evaluation of combination RTK and PI3K/Akt/mTOR pathway inhibition is warranted. In the trials above, there were occasionally patients who experienced exceptional responses from PI3K/Akt/mTOR pathway inhibition [113]. These occurrences suggest that there may be a particular subset of patients who will receive substantial benefit from inhibition of this pathway. The challenge remains in identifying who those exceptional responders will be. Attempts have been made with next-generation sequencing to identify mechanistic explanations for responders and non-responders, with mixed results [113,127]. More sophisticated molecular matching in an individual’s tumor’s genetic makeup may further improve outcomes [127,128].

## 5. MAPK Pathway

The mitogen-activated protein kinase (MAPK) signaling pathway is involved in cellular proliferation, growth, and survival [129]. In general, the activation of the MAPK pathway begins with growth factors binding their associated receptors, i.e., epithelial growth factor binding EGFR [130]. The tyrosine kinase receptor binding leads to activation of Ras GTPase (Ras), which is located intracellularly [131]. An activation cascade ensues, with RAS subsequently activating RAF, which then activates mitogen-activated protein kinase (MEK), which then activates MAPK [131,132]. Of note, there are 3 MAPKs, each with multiple isoforms (ERKs, JNKs, and p38-MAPKs). While there is significant overlap in their role/function, the ERKs are typically activated in response to growth stimuli, while JNKs and p38-MAPKs are activated in response to cellular stress and can have both anti- and pro-apoptotic effects [133]. By way of multiple downstream effectors, including c-Myc and NF-ĸB, MAPK is involved in the regulation of translation, differentiation, and the cell cycle [131,132]. There is frequent and complex cross-talk with several other intracellular pathways including the PI3K/Akt/mTOR pathway [101]. Genetic alterations in the MAPK pathway have been implicated in a wide array of malignancies including urothelial carcinoma [131]. In urothelial carcinoma, genetic alterations in RAS have been implicated in 2–5% of cases [31,134]. BRAF mutations have been noted in 2% of cases [134].

### MAPK Pathway Inhibitors

While there is a lower frequency of MAPK pathway genetic alterations in urothelial carcinoma than many of the potential therapeutic pathways already discussed, its interaction with more frequently mutated pathways and receptors lends it clinical relevance. Therapies directed at inhibition of the MAPK pathway have had mixed results. Tipifarnib is a farnesyltransferase inhibitor that has been shown to inhibit RAS function [135]. Tipifarnib was evaluated in a phase II clinical trial in patients with metastatic urothelial carcinomas with HRAS mutations [136]. Of the 21 patients evaluated, 19% had progression-free survival at 6 months, suggesting poor activity.

Sorafenib is an inhibitor with activity against multiple kinases including RAF [137]. Sorafenib as a single agent was found to have minimal activity in a phase II trial of patients with advanced urothelial carcinoma [138]. A phase I clinical trial of the combination of sorafenib and vinflunine was notable for an overall response rate of 41% in patients with post-platinum metastatic UC [139]. A phase II trial assessed sorafenib in combination with gemcitabine and carboplatin as first-line therapy in metastatic or unresectable UC [140]. While this study reports a median progression-free survival of 9.5 months, there were significant toxicities with the regimen leading to 65% of the cohort discontinuing treatment [140]. Further evaluation with phase III studies of sorafenib in combination with vinflunine or gemcitabine/carboplatin is warranted. In addition, patient selection based on genetic alterations in RAF may further augment results. The inhibition of the MAPK pathway may have a clinically relevant role, especially in combination with chemotherapy or when combined with additional targeted therapy.

## 6. Chromatin Remodeling

One of the striking findings in the Cancer Genome Atlas’ analysis of muscle-invasive bladder cancer was the frequency of genetic alterations in chromatin regulatory genes [31]. Urothelial carcinoma had some of the highest rates of alterations in chromatin regulatory genes of any cancer type. Dysregulation of chromatin remodeling was present in 89% of the samples. Some of the most commonly affected genes were MLL2 (27%), ARID1 (25%), KDM6A (24%), and EP300 (15%). The regulation of chromatin is an attractive therapeutic target because of its relevance to cellular plasticity, differentiation, DNA repair, transcription regulation, and many other nuclear functions [141]. However the specific pathogenetic implications of these mutations have not yet been sufficiently studied.

### Therapies Directed at Chromatin Remodeling

The relative frequency of these mutations does increase the potential yield of targeted therapies. Some hypothesize that genetic alterations in chromatin regulation may create an opportunity to exploit synthetic lethality by the targeted inhibition of additional chromatin regulators [141]. Synthetic lethality occurs when the simultaneous disruption of two genes/proteins leads to cellular death only when used in combination [142]. This principle can be exploited in cancer cells that already carry genetic alterations by addition of a synthetically lethal inhibitor. For instance, synthetic lethality has been demonstrated in ARID1A mutated cancers by inhibition of EZH2 methyltransferase in vivo [143]. Therapies such as histone deacetylase inhibitors (HDACi) have been suggested and are currently being evaluated [144]. The HDACi, vorinostat, was previously evaluated in urothelial carcinoma as a single agent and was found to be limited by toxicities [145]. Vorinostat is now being evaluated in conjunction with pembrolizumab in patients with advanced UC (NCT02619253). Mocetinostat was evaluated as a single agent in a phase II trial of patients with advanced/metastatic urothelial carcinoma and a genetic alteration in genes involved in chromatin remodeling, CREBBP, or EP300 [146]. Significant toxicities were noted leading to treatment interruptions and dose reductions. The study was terminated early due to lack of efficacy. The HDACi, belinostat, is currently being evaluated in combination with tremelimumab and durvalumab in patients with ARID1A mutations and unresectable, metastatic, or locally advanced UC (NCT05154994). Tazemeostat, an EZH2 inhibitor, is being evaluated in combination with pembrolizumab in a phase I/II trial of patients with locally advanced/metastatic urothelial carcinoma who have progressed on platinum chemotherapy or are platinum-ineligible (NCT03854474). Finally, as DNA repair can be inhibited by alterations in chromatin regulatory genes, a possible therapeutic strategy is to inhibit DNA repair with agents such as PARP inhibitors [141,147]. The evaluation of PARP inhibitors in MIBC is discussed later. All in all, chromatin remodeling is a promising therapeutic target in the management of MIBC.

## 7. Cell Cycle Regulation

The regulation of the cell cycle is another relevant target in bladder cancer. CDKN2A/B, the gene for p14 and p16, which inhibit the cyclin-dependent kinases, Cdk4 and Cdk6, is altered in 5–23% of cases of MIBC [31,38]. CCND1 (10–14% genetic alterations) and CCND3 (4–11% genetic alterations) encode for cyclins that promote cell cycle progression via interactions with Cdk4 and Cdk6. CDKN1A, a potent cyclin-dependent kinase inhibitor, is genetically altered in 14% of cases. The tumor suppressor RB1 is genetically altered in 13–17% of cases, while Tp53 is genetically altered in 49–54% of cases. MDM2, an important regulator of Tp53, is genetically altered in 9–11% of cases. All in all, cell cycle regulation is genetically altered in up to 93% of cases of muscle-invasive bladder cancer [31].

### CDK Inhibitors

CDK4/6 inhibition has been a therapeutic target in many solid cancers including urothelial carcinoma. Palbociclib, a CDK4/6 inhibitor was evaluated in a phase II trial of patients with metastatic, platinum-refractory urothelial carcinoma with loss of p16 and intact Rb. Unfortunately, Palbociclib was not effective; only 17% of patients achieved progression-free survival at 4 months [148]. In an analysis of mechanisms of resistance to CDK 4/6 inhibition, Tong et al. demonstrated alterations in DNA repair pathways are likely involved [149]. They have subsequently demonstrated the effectiveness of the combination of CDK 4/6 inhibition with a PARP inhibitor in pre-clinical studies [150]. This warrants further clinical study. The combination of abemaciclib, a CDK4/6 inhibitor, and CRISPR knockout of CDKN2A is being evaluated in a phase I trial in the neoadjuvant setting in platinum-ineligible patients(NCT03837821) [151]. Trilaciclib, another CDK 4/6 inhibitor, is being evaluated in patients with advanced/metastatic UC receiving chemotherapy followed by avelumab, which is predicated to improve outcomes by protection from myelosuppression of chemotherapy as well as enhance the immune tumor microenvironment by inducing a transient G1 cell cycle arrest of hematopoietic cells (NCT04887831).

## 8. DNA Damage Repair

Genetic alterations in DNA damage response (DDR) genes are also prevalent in bladder cancer. ERCC1 and ERCC2, which are components of the nucleotide excision repair pathway, are implicated [31,152]. Additional DDR genetic alterations include BRCA1/2 (6% and 14%, respectively), ATM (12%), RB1 (13–17%), and FANCC (2%) [31]. In general, DDR genetic alterations have been associated with improved responses to chemotherapy and immune checkpoint inhibitors, as well as improved responses to immune checkpoint inhibitors [4,153,154,155,156].

### Targeting Alterations in DNA Damage Repair

Targeted therapy towards urothelial carcinoma with DDR genetic alterations includes the usage of poly ADP-ribose polymerase (PARP) inhibitors. PARP enzymes are involved in repairing single-strand and double-strand DNA breaks. PARP inhibitors take advantage of the concept of synthetic lethality in tumor cells that have deficiencies in DNA repair [157].

Relevant PARP inhibitors include rucaparib, olaparib, niraparib, and talazoparib. Rucaparib was evaluated in patients with previously treated advanced or metastatic urothelial carcinoma, regardless of tumor homologous recombination deficiency (HRD) status [158]. Of the 97 patients enrolled, there were no confirmed responses. The study was terminated for lack of efficacy. However, patient selection for HRD status may have been suboptimal based on genome-wide loss of heterozygosity (LOH).

Olaparib is also being evaluated in combination with the immune checkpoint inhibitor durvalumab. A phase II trial in the neoadjuvant setting demonstrated tolerability of the regimen as well as efficacy, with a 50% pathologic complete response rate at the time of cystectomy [159]. The combination was evaluated in patients with advanced disease who had progressed on platinum-based chemotherapy in a multi-arm, multi-agent trial (BISCAY) [64]. The patients with and without DNA homologous recombination repair deficiencies were evaluated in separate treatment arms. The combination did not meet the efficacy criteria for continuation. Olaparib was also evaluated in combination with durvalumab in a phase II randomized trial in patients with metastatic UC who were platinum-chemotherapy-ineligible and had not previously received chemotherapy for stage IV disease [160]. A total of 154 patients were randomized to receive durvalumab with placebo or durvalumab with olaparib. The median PFS was not significantly different for durvalumab and placebo vs. durvalumab and olaparib. However, in a pre-specified subset of patients with homologous recombination repair (HRR) mutations, the progression-free survival was significantly improved in the group receiving olaparib and durvalumab (5.6 months vs. 1.8 months). Another randomized phase II trial demonstrated a trend for improved PFS with switch maintenance rucaparib in those with stable or responding disease on platinum-based chemotherapy [161]. The observed benefit in the subset of patients with HRR mutations confirms the importance of using biomarkers to select patients in subsequent evaluations of PARP inhibitors.

Olaparib is also being evaluated in a phase II study of patients with advanced or metastatic UC who have progressed on chemotherapy or immune checkpoint inhibitors and have genetic alterations in several DDRs including BRCA1, BRCA2, ATM, MSH2, PALB2, BRIP1, and many others (NCT03375307). PARP inhibitors in this setting are also included in two additional ongoing umbrella studies (NCT03682289, NCT03869190, NCT03992131).

PARP inhibition has also been evaluated as maintenance therapy in the metastatic setting. Niraparib was evaluated as maintenance therapy in metastatic UC after patients achieved an objective response (OR) or stable disease (SD) after receipt of first-line platinum-based chemotherapy [162]. Of the 58 patients enrolled, 39 were randomized to niraparib, and 19 were randomized to placebo. There was no difference between arms in progression-free survival. This study did not require DDR genetic alterations to be present for enrollment. While there was no improvement in PFS in the 21 patients with HRR genetic abnormalities, it is possible that a larger sample size of patients with DDR genetic alterations would be necessary to observe an effect. Rucaparib was evaluated in a phase II trial as maintenance therapy following platinum-based chemotherapy in patients with metastatic disease that exhibits DNA repair deficiency [161]. There was an improvement in PFS in patients receiving rucaparib vs. placebo (Median PFS 35.3 weeks vs. 15.1 weeks, respectively; HR 0.53, 80% CI 0.3–0.92). These results suggest that PARP inhibition may be more effective in patients whose disease is responsive to platinum-based therapies. Additional clinical trials are ongoing that evaluate PARP inhibitors as maintenance therapy in the metastatic setting in combination with an immune checkpoint inhibitor or in combination with a receptor tyrosine kinase inhibitor (Table 2).

To this date, PARP inhibitors have not shown promising results in the treatment of urothelial carcinoma. It is possible that patient selection could be optimized to improve the results. Further work is needed to select patients with the highest chance of benefiting from PARP inhibitors. In addition, the ongoing studies assessing PARP inhibitors in novel clinical settings, such as maintenance therapy, may prove to be efficacious.

## 9. Emerging Therapeutic Targets and Future Directions

In addition to the genetic alterations in the pathways discussed above, there are additional emerging therapeutic targets. Multi-kinase inhibitors such as cabozantinib (inhibitor of VEGFR2, c-MET, RET) are being evaluated in multiple settings of UC [163,164]. Cabozantinib was evaluated in a phase II trial as a single agent in patients with metastatic platinum-refractory UC [165]. The objective response rate was 19%. There are several trials ongoing with cabozantinib in addition to PD1/L1 inhibitors in the advanced and metastatic setting (Table 2). Pembrolizumab with lenvatinib, a multikinase inhibitor, was evaluated in a phase III, randomized, double-blinded trial of patients with advanced/metastatic UC ineligible to receive cisplatin-expressing PD-L1 or ineligible to receive platinum-based chemotherapy regardless of PD-L1 status [166]. Among these patients, 218 were randomized to pembrolizumab and placebo, while 223 were randomized to pembrolizumab and lenvatinib. There was no difference in progression-free survival or overall survival. There were more grade 3–5 adverse events in the group receiving lenvatinib (50% vs. 27.9%). Of note, the results were possibly compromised by a high proportion of poor-performance-status patients and by not incorporating biomarkers for FGFR GAs or other RTK GAs relevant to lenvatinib. Of note, lenvatinib has been associated with significant cardiotoxicity including qt interval prolongation and cardiotoxicity [167,168]. Among patients with lenvatinib, 7% were noted to develop cardiac dysfunction, with 2% noting grade 3 or greater. In addition to multi-kinase inhibitors, berzosertib, an inhibitor of ataxia telangiectasia and Rad3-related (ATR) protein, was evaluated in addition to chemotherapy in patients with metastatic UC who had not yet received chemotherapy for metastatic disease [169]. Unfortunately, there was no improvement in progression-free survival.

As advances in the understanding of the pathogenic mechanisms involved in cancer continue, there will undoubtedly be novel approaches to target genetic alterations in urothelial carcinoma. Emerging evidence is adding to our understanding of the role that phenotypic plasticity, epigenetic reprogramming, cellular senescence, the tumor microenvironment, and polymorphic microbiomes have in pathogenesis [170,171]. We anticipate attempts to target the recently appreciated pillars of pathogenesis will be forthcoming.

In recent years, there has been a welcomed increase in the availability of advanced and increasingly comprehensive genetic analysis of tumors. In tandem, there is also a growing appreciation of the genetic complexity of tumors; genetic alterations can be heterogeneously distributed within a tumor, especially in chemotherapy resistant tumors [172]. Refining patient selection in clinical trials is important to improving outcomes of targeted therapies in urothelial carcinoma. At a minimum, we suggest that patients evaluated in clinical trials for targeted therapies should have tumors with relevant genetic alterations. However, the presence of a genetic alteration alone does not suggest pathogenic relevance. The development of strategies to identify key pathogenic mutations more precisely in an individual tumor are needed to improve outcomes with targeted therapy.

## 10. Conclusions

The management of muscle-invasive and advanced bladder cancer has changed dramatically in the past five years. As the genetic alterations key to the pathogenesis of MIBC are identified at an accelerating pace, the potential for breakthroughs accelerates as well. From tyrosine kinase receptors to intracellular pathways to intranuclear processes, targeted therapies are being developed. At this point, targeting FGFR with erdafitinib is the most effective targeted therapy. There remains much to be learned about how erdafitinib can best be employed in the management of MIBC. There are also several other FGFR inhibitors being evaluated in various treatment settings and combination therapies. Meanwhile, targeting the ErbB receptors, PI3K/Akt/mTOR, MAPK, chromatin remodeling, cell cycle regulation, and DNA damage repair pathways have had mixed results. Despite this, there remains reason for optimism that with advanced molecular characterization of tumors, patient selection can continue to improve so that additional targeted therapies will be identified to further improve patient outcomes.

## Figures and Tables

**Figure 1 cancers-14-01795-f001:**
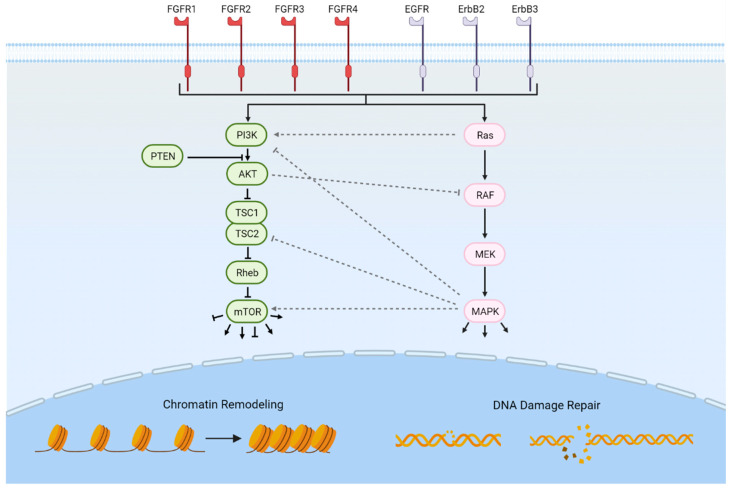
Potentially targetable genetically altered pathways in advanced bladder cancer. Frequently mutated pathways in advanced bladder cancer include RTKs such as the FGFRs and ErbB receptor groups. Intracellular pathways include the PI3K/Akt/mTOR pathway, the RAS-MAPK signaling pathway, chromatin remodeling, and DNA damage repair. Significant cross-talk exists between the PI3K/Akt/mTOR and RAS-MAPK pathways. Downstream effects of mTOR activation include cell cycle progression and cellular growth; downstream effects of MAPK activation include the regulation of translation, differentiation, and the cell cycle. Figure created with biorender.com (accessed on 22 February 2022).

**Table 1 cancers-14-01795-t001:** Ongoing Clinical Trials of FGFR Inhibitors in advanced or metastatic bladder cancer.

Drug Name	Phase	Setting	FGFR Status	Intervention	# of Pts	Status	NCT #
**AZD4547**	Ib	Metastatic, 2nd or 3rd line	FGFR1–4 GA	AZD4547, AZD4547 + durvalumab	156	Active, not recruiting	02546661
	II	Metastatic, progression through standard therapy	FGFR1–3 mutation/translocation	AZD4547	6452	Recruiting	02465060 *
**Derazantinib**	I/II	Advanced/Metastatic	FGFR1/2/3 GA	derazantinib vs. derazantinib + atezolizumab (multiple subgroups for first vs. second line as well as derazantinib dosing)	272	Recruiting	04045613
**Erdafitinib**	Ib	Metastatic, progression on platinum and PD1/L1	FGFR2/3 GA	erdafitinib + enfortumab vedotin	30	Recruiting	04963153
	II	Metastatic, progression through standard therapy	FGFR amp, mutation, or fusion	erdafitinib	6452	Recruiting	02465060 *
	I/II	Advanced/Metastatic	Not required	erdafitinib vs. erdafitinib + cetrelimab	126	Recruiting	03473743
	II	Metastatic	Not required	erdafitinib (intermittent vs. continuous dosing)	236	Recruiting	02365597
	III	Advanced/Metastatic	FGFR2/3 GA	erdafitinb vs. vinflunine or docetaxel vs. pembrolizumab	631	Recruiting	03390504
**Futibatinib**	II	Advanced/Metastatic—first line	+/−FGFR 1–4 GA	futibatinib + pembrolizumab	46	Recruiting	04601857
**Infigratinib**	I	Neoadjuvant, cisplatin-ineligible	FGFR2/3 GAs	infigratinib	12	Not yet recruiting	04972253
	III	Adjuvant	FGFR3 GA	infigratinib vs. placebo	218	Recruiting	04197986
**Lenvatinib**	III	Advanced/Metastatic—first line	Not required	pembrolizumab + lenvatinib vs. pembrolizumab	487	Active, not recruiting	03898180
**Pemigatinib**	II	Adjuvant, pT3–4 or pN1–3		pemigatinib	2	Active, not recruiting	04294277
	II	Advanced/Metastatic, progressed on first line	FGFR1–4 GAs	pemigatinib intermittent dose vs. pemigatinib continuous dose	263	Active, not recruiting	02872714
**Regorafenib**	II	Progression on all standard therapies	FGFR 1–4 GA	regorafenib	160	Recruiting	02795156
**Rogaratinib**	Ib/II	Advanced/Metastatic—first line, cisplatin-ineligible	High FGFR1/3 mRNA levels	rogaratinib + atezolizumab vs. atezolizumab	210	Active, not recruiting	03473756

* Trial with additional treatment arms not mentioned in the table. Abbreviations: GA, Genomic Alterations; FGFR, Fibroblast Growth Factor Receptor; NCT, National Clinical Trial.

**Table 2 cancers-14-01795-t002:** Ongoing clinical trials targeting genetic alterations in advanced bladder cancer (not including FGFRs).

Drug Name	Phase	Setting	Targeted GA	Intervention	#Pts	Status	NCT #
**ErbB Inhibitors**							
**Ado-Trastuzumab**	II	Adv/Met UC, Erbb2 GA	Erbb2	ado-trastuzumab	135	Recruiting	02675829
**Afatinib**	II	Adv/Met UC, no standard options available	Erbb1	afatinib	160	Recruiting	02795156 *
	II	Adv/Met UC, platinum-refractory	Erbb1	afatinib	95	Recruiting	02122172
**RC48-ADC**	II	Adv/Met UC, cisplatin-refractory ErbB2 overexpression	ErbB2	RC48-ADC	60	Active, not recruiting	03809013
**Tucatinib, Trastuzumab**	II	Adv/Met UC, Erbb2 GA	Erbb2	tucatinib + trastuzumab	270	Recruiting	04579380
**PI3K/Akt/mTOR Pathway**							
**Sapanisertib**	II	Adv/Met UC, platinum-refractory, with TSC GA	mTOR	sapanisertib	209	Active, not recruiting	03047213
**Temsirolimus** **Bevacizumab Cetuximab**	I	Adv/Met UC, relapsed after standard therapy	mTOR, EGFR, VEGF	temsirolimus + bevacizumab +/− cetuximab	155	Active, not recruiting	01552434
**Chromatin Remodeling**							
**Belinostat**	I	Adv/Met UC, no standard therapy available, ARID1A lof	HDAC	tremelimumab + durvalumab + belinostat	9	Recruiting	05154994
**Entinostat**	II	MIBC, Neoadjuvant	HDAC	entinostat + pembrolizumab	20	Recruiting	03978624
**Tazemetostat**	I/II	Adv/Met UC, cisplatin-refractory	EZH2	tazemetostat + pembrolizumab	30	Recruiting	03854474
**Vorinostat**	I/Ib	Adv/Met UC, platinum-refractory	HDAC	vorinostat + pembrolizumab	57	Active, not recruiting	02619253
**Cell Cycle Regulation**							
**Trilaciclib**	II	Adv/Met UC—first line	CDK4/6	trilaciclib + platinum chemo then trilaciclib + avelumab	90	Recruiting	04887831
**DNA Damage Repair**							
**Niraparib**	I/II	Adv/Met UC, platinum-refractory	DDR, RTKs	niraparib + cabozantinib	20	Recruiting	03425201
**Olaparib**	II	Adv/Met UC, platinum, or PD1/L1 refractory	DDR	olaparib	60	Recruiting	03375307
	II	Adv/Met UC, ARID1A mutated, ATM mutated	DDR, ATM	olaparib + AZD6738	68	Recruiting	03682289
**Talazoparib**	II	Adv/Met UC, s/p platinum with stable disease	DDR	talazoparib + avelumab	50	Recruiting	04678362
**Umbrella/Basket/Multi-Arm Trials**							
**BISCAY**	Ib	Metastatic UC, 2nd or 3rd line	HRR, CKDN2A, RB1, MAPK, mTOR	olaparib + durvalumab AZD1775 + durvalumab selumetinib + durvalumab vistusertib + durvalumab	156	Active, not recruiting	02546661 *
**My Pathway**	IIa	Adv/Met UC, Erbb1 or Erbb2 GA	Erbb1, Erbb2, BRAF	trastuzumab + pertuzumab or Erlotinib or vemurafenib + cobimetinib	676	Active, not recruiting	02091141 *
**The Match Screening Trial**	II	Adv/Met UC, relapsed after standard therapy	P13K, PTEN, AKT, mTOR, BRAF, NRAS, CCND1	copanlisib or capivasertib or ipatasertib or sapanisertib or trametinib or binimetinib or dabrafenib or palbociclib	6452	Recruiting	02465060 *
**SEASTAR**	Ib/II	Adv/Met UC progressed on 1 previous therapy	DDR, VEGF, FGFRs, PDGFRs	rucaparib + lucitanib rucaparib + sacituzumab govitecan	329	Active, not recruiting	03992131

* Trial with additional treatment arms not mentioned in the table. Abbreviations: Adv, Advanced; Met, Metastatic; TSC, Tuberous Sclerosis Complex; HRR, Homologous Recombination Repair; lof, loss of function; GA, Genetic Alteration; Pts, Patients; DDR, DNA Damage Repair; RTKs, Receptor Tyrosine Kinases; PD1/L1, Programmed Death 1/Programmed Death Ligand 1 Inhibitor; ATM, PDGR, Platelet-derived Growth Factor Receptors.

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
