# Peer review of "Molecularly Targeted Therapy towards Genetic Alterations in Advanced Bladder Cancer"

_cancers, 2022, doi:10.3390/cancers14071795_

Round 1

Reviewer 1 Report

This manuscript covers a timely and very complete review of targeted therapies in advanced bladder cancer. I can recommend its publication in this journal, with no major changes. Nonetheless, I encourage the authors to make a further revision of the text for minor corrections.

Minor points

The text needs revision for minor corrections. For example see lines 37 (in 2022), 122 (erdaftinib), 186-188, 216 (HER3), 395 (TMB), 592-593, 606 (parp), and references 31, 52, and 55.

Author Response

We greatly appreciate your feedback as well as the minor points highlighted. We have made the corrections to the errors which were noted by the reviewer. Including lines 37 (in 2022), 122 (erdaftinib), 186-188, 216 (HER3), 395 (TMB), 592-593, 606 (parp), and references 31, 52, and 55.

Reviewer 2 Report

The manuscript of Thomas and Sonpavde reviews thoroughly targeted therapies tested in Bladder cancers based on genetic alterations characteristic of this neoplasia.

The manuscript nicely covers this field and is well written.

I would suggest minor revisions:

-           Line 44: what is the median survival of patients not eligible for platinum-based chemotherapy? This information helps in the understanding of the improvement achieved with immunotherapy.

-           Line 80: is known the mechanism at the basis of FGFR upregulation?

-           Line 109-110: the authors should describe briefly the mechanism by which FGFR inhibitors exert their functions

-           Figure 1: it is not clear the function of mTOR and MAPK pathways. The authors should add more details to the figure.

-           Line 141: the authors should make uniform the writing of drugs (lower case/ upper case)

-           Lines 153-156: in my opinion, the sentence is unclear. Rephrase

-           Lines 186-188: the sentence seems incomplete

-           Lines 331-333: the author should briefly describe the rationale existing between PI3K alteration and ICIs treatment

-           Line 378: the authors should describe the connection existing between NF2 and PI3K

-           Lines 385-387: does the activation of AKT upon treatment with mTORC1 inhibitors may be due to a negative regulation on mTORC2 by mTORC1? Briefly describe

-           Line 456: Do the three different MAPKs (ERKs, JNKs, and p38 MAPKs) have the same role in the regulation of cellular functions?

Author Response

We greatly appreciate your feedback as well as the minor revisions suggested. Revisions, clarifications, and additions were made in the text as suggested. To summarize, the median survival of patients not eligible for platinum-based therapy was added. Of note regarding line 80, the basis of FGFR upregulation has not been clearly characterized in the literature.  The mechanisms of individual inhibitors are quite nuanced and beyond the scope of this article. A general statement was added. The caption of figure 1 was addended to provide further clarity. A rationale for PI3K alteration and ICI treatment was added. The role of mTORC2 in increasing the activation of AKT was added. The role of the three different MAPKs is quite nuanced and beyond the scope of the article. A summary sentence was added to provide further clarification.

Reviewer 3 Report

Reviewer’s report for Cancers-1643189

In their review article entitled "Molecularly Targeted Therapy Towards Genetic Alterations in Advanced Bladder Cancer", Thomas and Sonpavde discuss key genetic aberrations and summarize the current status of targeted therapies in muscle-invasive bladder cancer (MIBC).

Reviewer’s comments:

This is an interesting review, summarizing current targeted therapies against muscle-invasive bladder cancer, while also presenting results from a significant number of recent clinical trials.

There are no reservations regarding this article, since most of the recent therapies against muscle-invasive bladder cancer are discussed, while at the same time, significant information on current clinical trials is offered. Therefore, the paper may be published in Cancers after addition of the following:

  • PI3K/AKT and MAPK pathways interact with one another even when one of the two is inhibited. A relative nuance is made in Figure 1, but is not discussed in the text. Please, discuss briefly the PI3K/AKT-MAPK pathways cross-talk and its possible effects on the use of specific inhibitors against the first or the second pathway.
  • Please, provide an explanation on why chromatin remodeling could be an important target in MIBC therapy.
  • Explain what synthetic lethality is.
  • Add some information on cardiotoxicity which is induced by multiple kinase inhibitors or other targeted drugs.
  • Advanced tumors are genetically unstable and certainly consist of a variety of subclones. The same likely holds true for cisplatin resistant patients. What is therefore the meaning in treating the above patients with inhibitors against one or a few cellular targets? Please, discuss this in the final part of the manuscript.
  • Together with the above, please, make a more thorough discussion on why so many targeted drugs seem to perform poorly in a large number of UC clinical trials and present your opinion on the current design of clinical trials more strongly [what is the point in enrolling patients in clinical trials for targeted anticancer drugs when the molecular signature(s) of the tumor is unknown? Selection must make sure that patients indeed bear the specific molecular abnormalities against which they are being treated].
  • A suggestion would be to add more pathway explanatory figures in order to make the review easier to understand and more attractive.

Author Response

We greatly appreciate your feedback as well as the minor revisions suggested. Revisions, clarifications, and additions were made in the text as suggested. Regarding the note of  PI3K/AKT and MAPK cross-talk, this is discussed in paragraph 349-358.  An explanation of chromatin remodeling’s role as a therapeutic target is added. Synthetic lethality is defined. The cardiotoxic adverse events of Lenvatinib has been added. A paragraph in the second to last section was added to address the need to improve patient selection as an important next step to improve outcomes of targeted therapies in UC.